# Occurrence, Biosynthesis, and Health Benefits of Anthocyanins in Rice and Barley

**DOI:** 10.3390/ijms26136225

**Published:** 2025-06-27

**Authors:** Essam A. ElShamey, Xiaomeng Yang, Jiazhen Yang, Xiaoying Pu, Li’E Yang, Changjiao Ke, Yawen Zeng

**Affiliations:** 1Biotechnology and Germplasm Resources Research Institute, Yunnan Academy of Agricultural Sciences, Kunming 650205, China; 2Rice Research Department, Field Crops Research Institute, Agricultural Research Center, Cairo 12619, Egypt

**Keywords:** anthocyanins, rice, barley, biosynthesis, transcription factors, MYB, bHLH, WD40, health benefits, nutraceuticals

## Abstract

The occurrence of anthocyanins in rice (*Oryza sativa*) and barley (*Hordeum vulgare*) varies among cultivars, with pigmented varieties (e.g., black rice and purple barley) accumulating higher concentrations due to genetic and environmental factors. The biosynthesis of anthocyanins is regulated by a complex network of structural and regulatory genes. Key enzymes in the pathway include chalcone synthase (CHS), chalcone isomerase (CHI), flavanone 3-hydroxylase (F3H), dihydroflavonol 4-reductase (DFR), anthocyanidin synthase (ANS), and UDP-glucose flavonoid 3-O-glucosyltransferase (UFGT). These genes are tightly controlled by transcription factors (TFs) from the MYB, bHLH (basic helix–loop–helix), and WD40 repeat families, which form the MBW (*MYB-bHLH-WD40*) regulatory complex. In rice, *OsMYB* transcription factors such as *OsMYB3*, *OsC1*, and *OsPL* (Purple Leaf) interact with *OsbHLH* partners (e.g., *OsB1*, *OsB2*) to activate anthocyanin biosynthesis. Similarly, in barley, *HvMYB* genes (e.g., *HvMYB10*) coordinate with *HvbHLH* TFs to regulate pigment accumulation. Environmental cues, such as light, temperature, and nutrient availability, further modulate these TFs, influencing the production of anthocyanin. Understanding the genetic and molecular mechanisms behind the biosynthesis of anthocyanins in rice and barley provides opportunities for the development of biofortification strategies that enhance their nutritional value.

## 1. Introduction

Anthocyanins are secondary metabolites found in higher plants [1,2], contributing to coloration and protection against environmental stresses [3,4]. In cereals, pigmented varieties of rice (e.g., black, red, and purple) and barley (e.g., purple) are rich sources of anthocyanins [5,6,7]. Their consumption is linked to a reduced risk of chronic diseases such as cardiovascular disorders, diabetes, and cancer [8,9,10]. These natural pigments not only enhance the visual appeal of food but also contribute significantly to their nutritional and functional properties [11,12,13]. In recent years, anthocyanin-rich rice and barley varieties have attracted considerable attention due to their potential health benefits, positioning them as valuable functional foods in the global market [14,15,16].

Rice, a staple food for over half of the world’s population, exists in various pigmented forms, such as black, purple, and red rice, with black rice being renowned for its exceptionally high anthocyanin content [17,18]. Similarly, barley, an ancient cereal grain widely used in brewing and animal feed, also includes pigmented varieties such as purple and blue barley, which accumulate anthocyanins in their outer bran layers [19,20]. Unlike their non-pigmented counterparts (white rice and regular barley), these colored grains offer enhanced nutritional profiles, making them attractive for dietary interventions that aim to prevent chronic diseases [21]. The growing interest in anthocyanins stems from their potent antioxidant, anti-inflammatory, and anti-carcinogenic properties, which have been linked to a reduced risk of cardiovascular diseases, diabetes, obesity, and neurodegenerative disorders [22,23,24]. Additionally, anthocyanins exhibit challenges related to their stability due to their sensitivity to pH, temperature, light, and oxygen, prompting research into the processing and storage methods most able to preserve their bioactivity [25,26,27,28].

This review provides a comprehensive analysis of anthocyanins in rice and barley, covering the variation in their content among different cultivars, their chemical stability under different environmental and processing conditions, their compositional diversity (e.g., cyanidin-3-glucoside in black rice vs. acylated forms in barley), their biosynthetic pathways and genetic regulation, and their nutritional implications and health benefits, including their role in disease prevention. By understanding these aspects, researchers, food scientists, and nutritionists can better utilize anthocyanin-rich rice and barley in the development of functional foods, dietary supplements, and therapeutic applications, ultimately promoting healthier dietary choices worldwide.

## 2. Analysis of Anthocyanins Compositional Diversity, e.g., Cyanidin-3-Glucoside in Black Rice vs. Acylated Forms in Barley

Anthocyanins are secondary metabolites that belong to the flavonoid family and are primarily found in the outer layers of grains, fruits, and vegetables (Figure 1) show cyanidin-3-glucoside biosynthesis pathway. The anthocyanidin structure contains a flavylium cation (oxonium ion) form (positively charged oxygen), conjugated double bonds (responsible for color), and hydroxyl (^−^OH) and methoxy (^−^OCH_3_) groups that influence the stability and color of the plant. In rice and barley, anthocyanins are concentrated in the bran layer, contributing to their pigmentation and nutritional value [19,29,30], Table 1 reports the studies on analysis of the compositional diversity of anthocyanins in black rice and barley. Due to their health-promoting properties, there is growing interest in the breeding and processing of anthocyanin-rich cereal varieties.

### 2.1. Extraction and Quantification Methods

Regarding extraction methods as shown in Figure 2 [34], many methods can be used to determine and extract the anthocyanins present in cereals and grains such as rice and barley [35]; these include solvent-based extraction (methanol, ethanol, acidified solvents) [36], ultrasound-assisted extraction (UAE) and microwave-assisted extraction (MAE), which enhance yield and efficiency [37], and supercritical fluid extraction (SFE), with the use of CO_2_ emerging as an eco-friendly method [38].

For quantification and characterization, spectrophotometry (pH differential method) is able to rapidly determine the total anthocyanin content [39,40]. High-Performance Liquid Chromatography (HPLC) coupled with Diode Array Detection (DAD) or Mass Spectrometry (MS) enables the precise identification of individual anthocyanins [40,41]. UPLC-MS/MS offers higher resolution and faster analysis [41].

Anthocyanin-rich rice and barley offer significant health benefits and economic potential. Advances in analytical techniques have improved the precision of anthocyanin profiling, while innovative extraction methods have enhanced their yield and stability. Future research should focus on optimizing cultivation practices and developing value-added products in order to maximize the utilization of these bioactive compounds.

### 2.2. Anthocyanin Composition in Rice and Barley

Anthocyanins are found in various parts of plants, including grains such as rice and barley [42]. In rice, anthocyanins are typically located in the pericarp (outer layer) of the grain [43,44]. Some black and red rice varieties produce anthocyanins and proanthocyanidins in the pericarp [44,45]. The content and composition of anthocyanins can vary significantly among different colored rice varieties [46]. The content of anthocyanins in rice and barley differs according to the variety and the color of the grains, as shown in Table 2:

Black and purple rice varieties are rich in cyanidin-3-glucoside (C_3_G) and peonidin-3-glucoside (P_3_G) [50]. In rice (*Oryza sativa*), the anthocyanin content varies with genotype, ranging from 0.1 to 6 mg/g in pigmented rice [51]. Red rice contains proanthocyanidins rather than anthocyanins, contributing to its distinct color [51]. Regarding barley (*Hordeum vulgare*), purple barley contains delphinidin-3-glucoside, cyanidin-3-glucoside, and petunidin derivatives [52]. The anthocyanin levels in barley range from 0.05 to 2.5 mg/g, depending on the cultivar and growing conditions [53].

The color of rice and barley grains, ranging from white, red, and purple, to black, is primarily determined by the type and concentration of anthocyanins and other phenolic compounds present in their bran layers. The variation in anthocyanin profiles among different colored grains is influenced by multiple factors, including genetic, biochemical, and environmental determinants [54,55,56,57]. Below are the key reasons for these differences:

#### 2.2.1. Genetic Factors (Biosynthetic Pathway Regulation)

Genetic factors play a crucial role in regulating the biosynthetic anthocyanin pathway in rice and barley, influencing pigmentation and stress resistance. Key genes, such as MYB transcription factors, bHLH proteins, and WD40 repeat proteins, along with structural genes like CHS, DFR, and ANS, orchestrate anthocyanin production. While barley exhibits more natural anthocyanin accumulation (e.g., in purple grains), rice often requires genetic engineering or mutagenesis to enhance anthocyanin content. Understanding these genetic mechanisms enables the development of biofortified crops with improved nutritional and agronomic traits.

##### Differential Expressions of Anthocyanin Biosynthesis Genes

The color variation is largely controlled by the phenylpropanoid and flavonoid biosynthesis pathways, which produce different anthocyanin derivatives [58,59,60]. Some of the key regulatory genes include CHS, which initiates flavonoid synthesis, DFR, which determines the type of anthocyanidin (cyanidin, delphinidin, pelargonidin), ANS, which converts leucoanthocyanidins to colored anthocyanidins, and UFGT, which adds sugar moieties, thus stabilizing anthocyanins [60,61,62,63].

##### Presence or Absence of Specific Transcription Factors

Anthocyanins, which contribute to pigmentation and stress resistance in plants, are regulated by key transcription factors (TFs) such as MYB, bHLH, and WD40, forming the MBW complex. However, the presence and functionality of these TFs vary between monocots like rice and barley, influencing anthocyanin production. For rice generally lacks strong anthocyanin pigmentation in most tissues due to inactive or absent regulatory TFs. While rice has MYB (e.g., OsC1) and bHLH (e.g., OsB1, OsB2) genes, their expression is often weak or restricted to specific tissues (e.g., apiculus and pericarp). Some wild rice varieties (e.g., Oryza rufipogon) accumulate anthocyanins due to functional MYB activators, but cultivated rice typically lacks these. Also, barley shows more anthocyanin accumulation than rice, particularly in purple-grained varieties, regulated by MYB TFs (e.g., HvMYB1, HvMYB2) and bHLH partners. The Ant2 locus in barley is linked to purple pigmentation, driven by an active MYB TF. Unlike rice, barley has functional MBW complexes that enhance anthocyanin biosynthesis in grains and leaves under stress. The differences between them, rice has limited anthocyanin production due to low TF activity or repression, whereas barley exhibits stronger pigmentation due to active MYB/bHLH regulators. Genetic engineering in rice (e.g., introducing OsC1 + OsB2) can induce anthocyanins, while barley naturally activates these pathways. Transcription factors regulate anthocyanin production for example, in black rice, the high expression of OsMYB3 and OsC1 leads to the accumulation of cyanidin-3-glucoside [42,64,65]. In purple barley, HvMYB10 activates the production of delphinidin-based anthocyanins [66], as shown in Figure 3 [67] transcription factors and its role on anthocyanin production.

##### Mutations and Epigenetic Modifications

In rice, mutations in genes like *OsC1* (a MYB transcription factor) and *OsDFR* (dihydroflavonol 4-reductase) disrupt anthocyanin production, leading to colorless grains. Similarly, in barley, mutations in Ant2 (a bHLH transcription factor) or structural genes (*CHS*, *F3H*) reduce pigmentation. Such mutations are exploited in breeding to either enhance or suppress anthocyanin levels for nutritional or agronomic benefits. Epigenetic regulation through DNA methylation and histone modifications dynamically control anthocyanin genes. In rice, hypermethylation of promoter regions (e.g., in *OsANS*) can silence expression, while demethylation activates it. Barley exhibits stress-induced epigenetic changes that upregulate anthocyanin-related genes (*HvMYB10*, *HvGT1*), enhancing pigment accumulation under environmental cues. Some rice varieties lose pigmentation due to mutations in anthocyanin-related genes (e.g., *Kala4* in black rice) [29,68,69]. In addition, DNA methylation and histone modifications can silence anthocyanin biosynthesis genes, leading to white or light-colored grains [70,71].

##### Utilizing Genetic Engineering to Enhance Anthocyanin Content in Rice and Barley

Rice (*Oryza sativa*) and barley (*Hordeum vulgare*) are staple crops with a limited natural anthocyanin content, which is primarily found in pigmented varieties such as black rice and purple barley [72,73]. Genetic engineering (transcription factors) Figure 4 [44] offers a promising approach to enhancing the biosynthesis of anthocyanins in these grains, improving their nutritional and functional properties. Anthocyanins, flavonoid pigments with health-promoting properties, are attractive targets for metabolic engineering in cereals like rice and barley. However, regulating their biosynthetic pathway requires careful consideration to ensure optimal agronomic performance and consumer acceptance. The anthocyanin biosynthesis pathway involves key enzymes such as PAL, CHS, DFR, and ANS. In rice and barley, overexpression of transcription factors like MYB, bHLH, and WD40 can enhance anthocyanin production. For example, in rice the *OsC1* (MYB) and *OsB2* (bHLH) genes regulate purple pericarp pigmentation. Barley the *Ant2* (MYB) gene controls anthocyanin accumulation in grains. Precise regulation is crucial to avoid metabolic trade-offs that may affect plant growth. Strategies include tissue-specific promoters (e.g., endosperm-specific glutelin promoters in rice) to limit pigment production to edible parts, and inducible systems (stress-responsive promoters) to trigger anthocyanin synthesis without compromising yield.

Agronomic performance considerations excessive anthocyanin production can lead to reduce biomass due to resource diversion from growth and pleiotropic effects on stress responses and nutrient uptake. For consumer acceptance as an important part while pigmented rice (black rice) is traditionally accepted, barley with altered grain color may face market resistance. Strategies to improve acceptance include retaining familiar grain appearance (purple barley with minimal color change) and highlighting health benefits (antioxidant, anti-inflammatory properties). Balancing anthocyanin pathway regulation in rice and barley requires a multidisciplinary approach optimizing gene expression, ensuring field performance, and addressing consumer preferences. Advances in CRISPR and synthetic biology offer precise tools to achieve this balance, enhancing nutritional value without compromising agronomic traits [74,75].

##### Strategies for Enhancing Anthocyanin Production via Genetic Engineering

Genetic engineering offers promising strategies to enhance anthocyanin production in rice and barley, leveraging key biosynthetic pathway genes (e.g., *PAP1*, *MYB* transcription factors, *DFR*, *ANS*) and regulatory elements. Approaches such as overexpression of anthocyanin-related genes, CRISPR/Cas9-mediated gene editing, and modulation of competing pathways can significantly boost anthocyanin accumulation. Additionally, tissue-specific promoters and stress-inducible systems can optimize production without compromising plant growth. While challenges like metabolic burden and regulatory approval remain, advances in genetic engineering hold great potential for developing nutrient-enriched rice and barley varieties with enhanced health benefits.

##### Over Expression of Key Anthocyanin Biosynthesis Genes

Anthocyanin biosynthesis is a well-studied metabolic pathway that involves the coordinated action of structural genes (encoding enzymes) and regulatory genes (transcription factors) [63,76,77]. These components work together to control the production, accumulation, and distribution of anthocyanins in plants. Understanding their roles is essential for genetic engineering strategies that aim to enhance the content of anthocyanins in crops such as rice and barley. Regarding the structural genes involved in anthocyanin biosynthesis, the anthocyanin pathway is a branch of the flavonoid pathway, starting from phenylalanine and leading to the formation of colored pigments [62,78,79]. Regarding the regulatory genes that control anthocyanin biosynthesis, while structural genes encode the enzymes, TFs regulate their expression; the most well-known regulatory system is the MBW complex [80,81,82].

Overexpressing anthocyanin biosynthesis genes in rice and barley can enhance nutritional value and stress resistance. There are many keys candidate genes including structural genes; PAL was the first enzyme in the phenylpropanoid pathway, CHS and CHI were Catalyze early steps in flavonoid biosynthesis, and DFR, ANS, and UFGT were essential for anthocyanin production. The other genes were regulatory genes (Transcription Factors). The key Overexpression targets in rice included *OsANS*, *OsDFR*, and *OsMYB* genes have been successfully overexpressed, leading to purple endosperm or leaves. Barley has *HvANT2* (anthocyanin activator) and *HvMYB10* overexpression enhances pigmentation in grains. While overexpression can increase anthocyanin content, unintended metabolic shifts or yield penalties may occur. Precision breeding (CRISPR/Cas9) or tissue-specific promoters can optimize anthocyanin production without compromising agronomic traits [83,84,85]. 

##### Challenges in Engineering Anthocyanin Biosynthesis

There are many challenges associated with the engineering of anthocyanin biosynthesis; in competing pathways, flavonols and proanthocyanidins share precursors, while suppressing *FLS* or *LAR* may increase the production of anthocyanins [86,87,88]. Tissue-specific accumulation, which constitutes overexpression, may reduce yield; therefore, endosperm-specific promoters (e.g., *Glutelin* in rice) are preferred [89,90,91]. After post-translational modification, anthocyanins degrade easily; therefore, introducing *AATs* or *GTs* from other species could improve their stability [92,93].

Anthocyanin biosynthesis is tightly controlled by structural genes (enzymes) and regulatory genes (*MYB,*
*bHLH*, *WD40*) [64,94,95]. Genetic engineering could enhance the production of anthocyanins in rice and barley by overexpressing key enzymes (*DFR*, *ANS*, *UFGT*) [63,96], introducing or activating transcription factors (*MYB*/*bHLH*), and blocking competing pathways (CRISPR knockout of *FLS*/*LAR*). Future work should focus on tissue-specific expression and metabolic balancing to maximize the content of anthocyanins without compromising their agronomic traits.

There is growing interest in enhancing anthocyanin production in crops through genetic engineering. However, several challenges hinder the efficient manipulation of anthocyanin biosynthesis. This review explores these challenges, including pathway complexity and regulatory mechanisms. Anthocyanin biosynthesis is a branch of the flavonoid pathway, involving multiple enzymatic steps regulated by a network of genes. There are some key enzymes including PAL, CHS, CHI, F3H, DFR, ANS, and UFGT. The Challenges are multi-enzyme coordination through engineering. A single gene may not guarantee increased anthocyanin production due to rate-limiting steps and feedback inhibition. Tissue-specific expressions in which anthocyanins are often produced in specific tissues (e.g., fruit skins, flower petals), requiring precise spatial and temporal regulation. Post-translational modifications which glycosylation and methylation of anthocyanins affect their stability and color, adding another layer of complexity.

For regulatory mechanisms and transcription factor interactions, anthocyanin biosynthesis is tightly regulated by transcription factors (TFs) and has some challenges; combinatorial control, the MBW complex (*MYB-bHLH-WD40*) must be precisely regulated; overexpression of one component may not suffice. Species-specific regulation, TFs from one plant may not function optimally in another due to differences in promoter recognition or protein interactions. Pleiotropic effects, Overexpressing anthocyanin regulators can inadvertently affect other metabolic pathways, leading to unintended phenotypes (e.g., stunted growth). To overcome these challenges, strategies include CRISPR-Cas9-mediated precision editing to fine-tune anthocyanin regulators, synthetic biology approaches (e.g., artificial gene clusters) for coordinated expression, and metabolic engineering in microbial hosts (*E. coli* and yeast) for scalable anthocyanin production. While genetic engineering of anthocyanin biosynthesis holds great promise, the pathway’s complexity, regulatory constraints, and stability issues present significant hurdles. Advances in systems biology, gene editing, and synthetic biology will be crucial in developing robust strategies for enhancing anthocyanin production in crops and microbial systems. Addressing ethical and regulatory concerns will also be essential for the successful deployment of engineered anthocyanin-rich products.

#### 2.2.2. Biochemical Composition of Anthocyanins

Rice and barley cereals exhibit variations in anthocyanin profiles due to genetic and environmental factors. Understanding their biochemical composition enhances their potential use in functional foods and nutraceuticals for improved human health

##### Type of Anthocyanidins

According to the structure of aglycone, there are many types of anthocyanidins as shown in Figure 5 [97]: cyanidin-based (red purple) anthocyanidins are common in red and purple rice/barley, delphinidin-based (blue purple) anthocyanidins are found deep in purple/blue barley, peonidin-based (red-magenta) anthocyanidins are present in some black rice varieties, and pelargonidin-based (orange red) anthocyanidins are rare in cereals but found in some mutant lines [98,99,100,101].

##### Glycosylation and Acylation Patterns

Anthocyanins differ in their sugar attachments (e.g., glucoside, rutinoside) and acylation (e.g., sinapic, malonic acid). For example, cyanidin-3-glucoside (C_3_G) dominates in black rice, while delphinidin-3-rutinoside appears in purple barley [102,103].

##### Co-Pigmentation with Other Flavonoids

The interaction of anthocyanins with flavones, flavonols, and proanthocyanidins alters their color stability and hue. Proanthocyanidins (condensed tannins) contribute to the red color of rice, rather than anthocyanins [104,105,106].

#### 2.2.3. Environmental and Agronomic Influences

##### Light Exposure (UV Radiation)

Anthocyanin synthesis involves the exposure of light-dependent grains to greater UV light, leading to the accumulation of more pigments [82,107,108]. Shaded conditions reduce the content of anthocyanins, leading to lighter grains [109].

##### Temperature Stress

Temperature stress affects the content and composition of anthocyanins in grains, with cold stress enhancing the production of anthocyanins as a protective mechanism [110,111]. High temperatures may degrade anthocyanins, reducing the intensity of their color [92,112,113,114].

##### Soil Nutrients, pH, and Post-Harvest Processing

Soil nutrients affect the content of anthocyanins in cereal grains. For example, nitrogen deficiency increases the synthesis of anthocyanins as a stress response [115]; a low soil pH (acidic conditions) enhances the stability of anthocyanins (red/purple hues) [92,116], and high levels of phosphorus may suppress the accumulation of anthocyanins [117,118]. During post-harvest processing, milling removes the bran layer in which anthocyanins are concentrated, turning purple/black rice into white rice. In addition, fermentation and thermal processing can degrade anthocyanins, altering the color of the grain [119,120,121].

#### 2.2.4. Evolutionary and Ecological Adaptations

In evolutionary and ecological adaptations such as wild vs. cultivated varieties, wild rice/barley often has a higher content of anthocyanins as a defense against pests and UV damage [122,123]. As a geographical adaptation, Himalayan purple barley has a unique content of anthocyanins due to its exposure to high-altitude UV light [124].

The differences in the color of rice and barley grains are primarily detected Via the genetic regulation of anthocyanin biosynthesis pathways, the biochemical modification (glycosylation, acylation) of anthocyanins [125,126], environmental factors (light, temperature, soil conditions), and the effects of post-harvest processing [127,128]. Understanding these factors aids in breeding colored grain varieties with enhanced nutritional benefits and in developing stable natural food colorants. Future research should assess the application of CRISPR-based gene editing to enhance the production of anthocyanins, conduct metabolomic studies to identify novel anthocyanin derivatives, and perform climate-resilient breeding to maintain the stability of pigments under stress.

## 3. Analysis of Anthocyanins Chemical Stability Under Different Environmental and Processing Conditions

Anthocyanin compounds are not only important for plant pigmentation but also possess significant health benefits. However, their chemical stability is influenced by various factors such as pH, temperature, light, oxygen, and processing methods [92,129,130]. Understanding the stability of anthocyanins in rice and barley is crucial for food processing, storage, and nutritional retention. Many factors affect the stability of anthocyanins, as shown in Table 3:

### 3.1. pH

Anthocyanins exhibit structural changes depending on the pH, which directly affects their color and stability [131,132]. Anthocyanins are most stable under acidic conditions (pH 1–3), appearing red [133]. On the other hand, in neutral to slightly acidic (pH 4–6) conditions, they may turn colorless or purple due to the formation of quinoidal bases [134,135]. However, under alkaline conditions (pH > 7), degradation accelerates, leading to brownish hues. In cereal grains such as rice, black rice anthocyanins (mainly cyanidin-3-glucoside) are more stable at a low pH but degrade rapidly at a neutral or alkaline pH. The purple anthocyanins (primarily delphinidin and cyanidin derivatives) in barley show similar pH-dependent instability, with greater degradation at pH > 5. The structural changes that occur in anthocyanins due to changes in the pH are caused by shifts in protonation states, as well as alterations in the conjugation and electron distribution; this leads to different colored forms (red in acid, blue in alkali) and eventual degradation at an extreme pH [136,137]. In addition, the color and structure of anthocyanins change with pH due to protonation/deprotonation reactions, leading to different molecular forms; under strongly acidic conditions (pH < 2), the flavylium cation (red) dominates. The positively charged oxygen in the central ring stabilizes the structure. The highly conjugated system absorbs light in the visible range (~500–550 nm), producing red hues. Under mildly acidic to neutral (pH 3–6), the loss of a proton (H^+^) converts the flavylium cation into a carbinol pseudo-base (colorless) at pH 3–4; in addition, water attacks the flavylium cation, breaking the conjugation and resulting in a loss of color. Under chalcone (yellow, less stable), further deprotonation leads to ring opening, reducing the intensity of the color [138,139]. A neutral to alkaline (pH 7–8) quinoidal base (blue/purple) forms due to the deprotonation of hydroxyl groups, and extended conjugation shifts the absorption to longer wavelengths (~600 nm), causing blue/purple colors. However, this form is less stable and degrades faster. Under highly alkaline (pH > 8) conditions, further deprotonation leads to degradation and the quinoidal base breaks down into phenolic acids and aldehydes, causing browning and the loss of pigment [92,140].

### 3.2. Temperature

Temperature is considered one of the most important factors affecting the stability of anthocyanins; this is because high temperatures during cooking and processing can degrade anthocyanins. Temperature affects the structure of anthocyanins by inducing chemical degradation, altering molecular interactions, and shifting the equilibria between different forms [141,142]. For rice, low temperatures (10–20 °C) increase the content of anthocyanins in leaves and grains (cold stress response) [143] and upregulate *OsC1* (a MYB transcription factor). High temperatures (>30 °C) suppress anthocyanins due to degradation and the reduced expression of *OsMYB* [65,144]. At low temperatures, anthocyanins are strongly induced in the hulls and stems of barley (e.g., *HvANT1* activation) [145,146]; this is used as a marker for cold tolerance in breeding. High temperatures suppress the production of anthocyanins less in barley than in rice, with some purple barley retaining pigments even at 35 °C. Regarding the effect of high temperatures on rice, it has been observed that steam cooking reduces the anthocyanin content by 20–40%, while boiling leads to the leaching of anthocyanins into water. In barley, thermal processing (e.g., roasting, malting) decreases the content of anthocyanins, with its stability decreasing above 60 °C. Anthocyanins consist of an anthocyanidin (aglycone) backbone linked to sugar moieties (glycosides); high temperatures can hydrolyze these glycosidic bonds, leading to the formation of less stable aglycones. Prolonged or excessive heat can cause ring cleavage, oxidation, or polymerization, resulting in the loss of color (fading or browning). Increased temperatures can alter solubility, leading to the precipitation or polymerization of anthocyanins. In some cases, heating may cause self-association (stacking of anthocyanin molecules), which can temporarily stabilize color but may eventually lead to degradation [147,148].

### 3.3. Light and Oxygen Exposure

Anthocyanins are highly susceptible to oxidation and photodegradation. Storage in opaque, airtight containers aids in the preservation of anthocyanins in both rice and barley [25,149,150]. Regarding rice, milled black rice loses anthocyanins faster due to increased surface exposure. The same is true for barley, as whole-grain barley retains anthocyanins better than pearled barley due to the presence of protective bran layers. Both light and oxygen induce structural changes in anthocyanins, leading to color fading or browning through mechanisms such as photo-oxidation, ring cleavage, and enzymatic polymerization [50,151]. The rate of degradation depends upon the anthocyanin structure (hydroxylation vs. methoxylation), pH (an acidic pH is more stable), and presence of co-factors (metal ions, enzymes, antioxidants). Stabilization strategies include storage in dark, oxygen-free environments, the use of co-pigments (e.g., flavonoids), acidic conditions (pH < 3), and the addition of chelating agents (e.g., citric acid) to bind metal ions. It is crucial that food scientists and manufacturers understand these factors in order to preserve anthocyanin-rich products (like wines, jams, and natural colorants) [14,152]. The effect of light exposure and UV radiation on rice is as follows [153]:

The synthesis of anthocyanins in rice is strongly dependent upon light, particularly in pigmented varieties (e.g., black/purple rice) [154]. The mechanism involves the activation of the transcription factors (OsMYB, OsbHLH) that upregulate anthocyanin genes (OsANS, OsDFR) via the phytochrome and cryptochrome pathways [155]. For example, UV-B radiation enhances the accumulation of anthocyanins as a protective response [156]. Some rice cultivars (e.g., Jiaoyu 5) show higher anthocyanin levels under blue/UV light but reduced levels in low light [157].

On the other hand, the effect of light exposure and UV radiation on barley is as follows: anthocyanin synthesis is regulated by light but is less dependent on UV compared to rice [158]; blue light is more effective than UV in activating HvMYB and HvTTG1 (key regulators) [159]; and purple barley varieties (e.g., *Hordeum vulgare* L. var. nigrum) show stronger red-light induction than rice [160]. In addition, there is variability in the accumulation of anthocyanins in some barley genotypes even in shaded conditions, which does not occur in rice.

### 3.4. Enzymatic and Non-Enzymatic Degradation

Non-enzymatic degradation occurs due to chemical and physical factors, including pH changes, temperature, light, oxygen, and interactions with other compounds. The enzymes naturally present in plants or created via microbial activity can break down anthocyanins. The key enzymes involved include polyphenol oxidase (PPO) and peroxidase (POD), which can degrade anthocyanins [161,162]. In rice, soaking and fermentation may reduce the content of anthocyanins due to the presence of enzymatic activity [163,164]. In barley, germination (malting) can increase the activity of enzymes, leading to the breakdown of anthocyanins. Both enzymatic and non-enzymatic degradation pathways disrupt the structure of anthocyanins, primarily by breaking glycosidic bonds (leading to unstable aglycones), oxidizing phenolic rings (destroying the chromophore), and converting flavylium cations to colorless forms (pH-dependent) [165,166].

### 3.5. Water Stress (Drought vs. Waterlogging)

The effect of water stress on rice is as follows: drought stress increases the content of anthocyanins in leaves (ROS scavenging mechanism) [167], *OsMYB3* is induced under dehydration [153], and waterlogging decreases the content of anthocyanins due to hypoxia and a reduction in photosynthesis [154]. The effect of water stress on barley is as follows: drought stress leads to the greater induction of anthocyanins compared to rice (linked to *HvMYB10*) [153]; purple barley shows better drought resilience due to antioxidant protection [155]; and the effects of waterlogging are less prominent than those in rice (barley is more drought-adapted than flood-tolerant) [156].

### 3.6. Soil Nutrients (Nitrogen, Phosphorus, Metals)

Low nitrogen (N) levels in rice increase the content of anthocyanins (stress response), while high levels of nitrogen decreases pigmentation (dilution effect) [115,157]. Metal stress (Al, Cd) induces the production of anthocyanins in roots (detoxification mechanism) [158,159]. Low nitrogen levels in barley lead to the strong accumulation of anthocyanins in leaves (more than rice) [19,124], while phosphorus (P) deficiency increases the content of anthocyanins (barley shows higher P-stress response than rice) [160,168]. Metal stress affected the content of anthocyanins less in barley compared to rice (barley relies more on phytochelatins) [158,169,170].

### 3.7. Strategies to Enhance Anthocyanin Stability

Some forms of chemical processing can affect the stability of anthocyanin production in grains such as rice and barley:Acidic processing: using citric or ascorbic acid in food formulations can stabilize anthocyanins [132].Encapsulation: microencapsulation techniques protect anthocyanins from degradation [171].Cold storage: refrigeration slows down anthocyanin degradation in rice and barley products [172].Reduced oxygen packaging: vacuum or nitrogen-flushed packaging minimizes oxidation [173].

The anthocyanins present in rice and barley are highly bioactive but chemically unstable under certain conditions. Their stability depends on the pH, temperature, light, oxygen, and processing methods used. Proper storage, acidic environments, and encapsulation can aid in the preservation of these valuable pigments, ensuring that their health benefits are retained in food products. Further research on stabilization techniques could enhance the utilization of anthocyanin-rich rice and barley in functional foods and nutraceuticals.

**Table 3 ijms-26-06225-t003:** Analysis of the chemical stability of anthocyanins under different environmental and processing conditions.

Environmental Conditions		Rice	Barley	References
Water conditions	Drought	Moderate increase	Strong increase	[174,175]
	Water logging	Sharp decrease	Mild decrease	[174]
Nutrient Condition	Low N	Moderate increase	Strong increase	[174]
	Low P	Mild increase	Strong increase	[174,175]
	Heavy metals	High induction (roots)	Low induction	[175]
Environmental factor	Light dependency	High (UV/blue)	Moderate (blue/red)	[113]
	Cold response	Moderate (grain/leaf)	Strong (hull/stem)	[113,175]
	Heat tolerance	Low (degrades >30 °C)	Moderate (retains pigments)	[113,175,176]
	Drought induction	Moderate	Strong	[113,175]
	Nutrient stress	N and metal-sensitive	N and P-sensitive	[174]

## 4. Implications of the Nutritional and Health Benefits of Anthocyanins, and Their Role in Disease Prevention

Anthocyanins are a subgroup of flavonoids; these bioactive compounds have attracted significant attention due to their potent antioxidant properties and potential health benefits [10,48,174]. This review explores the nutritional implications of anthocyanins, focusing on their dietary sources, bioavailability, metabolism, and health-promoting effects.

### 4.1. Dietary Sources of Anthocyanins

Anthocyanins are abundant in various plant-based foods, as shown in Figure 6 [177,178,179,180,181,182]. These foods include:-Fruits: berries (blueberries, blackberries, strawberries, raspberries), cherries, grapes, pomegranates, and blackcurrants.-Vegetables: red cabbage, eggplant, purple sweet potatoes, and red onions.-Other sources: red wine, tea, and certain grains such as black rice and barley.

The concentration of anthocyanins varies depending on factors such as the cultivar, ripeness, storage conditions, and processing methods [92,183].

### 4.2. Bioavailability and Metabolism

Despite their health benefits, anthocyanins have relatively low bioavailability (less than 1% absorption in some cases) [184,185]. No official recommendations regarding the daily intake of anthocyanins exist, but studies suggest that 12.5–50 mg/day would have health benefits; for example, a serving of blueberries (1 cup) provides ~150–200 mg anthocyanins [186,187]. In addition, some key factors affect their absorption and metabolism. These include:-Chemical structure: glycosylation (sugar attachment) influences absorption [185,188].-Gut microbiota: intestinal bacteria metabolize anthocyanins into smaller phenolic acids, enhancing bioavailability [184,189].-Food matrix: the presence of fiber, fats, and other compounds can either inhibit or enhance absorption [184,190,191].-Processing methods: heating and fermentation may degrade anthocyanins but can also release bound forms, improving absorption [191,192,193].

After absorption, anthocyanins undergo phase II metabolism (glucuronidation, sulfation, methylation) in the liver before being distributed to tissues [194,195,196].

### 4.3. Health Benefits and Nutritional Implications

Epidemiological and experimental studies suggest that anthocyanins may reduce the risk of chronic diseases, including cardiovascular disease (CVD), diabetes, cancer, and neurodegenerative disorders. However, while some evidence supports these claims, critical gaps remain in bioavailability, mechanistic understanding, and clinical applicability. Additionally, the commercialization of anthocyanin-rich products often exaggerates benefits without sufficient scientific backing. This discussion evaluates the current evidence, methodological limitations, and potential overstatements in health claims surrounding anthocyanins.

-Antioxidant and anti-inflammatory effects

Anthocyanins exhibit strong free radical-scavenging activity; they reduce oxidative stress, a key factor in aging and chronic diseases, by neutralizing reactive oxygen species (ROS), enhancing endogenous antioxidant defenses (e.g., superoxide dismutase and glutathione), and thus protecting cells from damage [197,198]. Their anti-inflammatory properties are linked to the inhibition of pro-inflammatory cytokines (e.g., TNF-α, IL-6) and NF-κB signaling pathways, which may aid in the management of inflammatory conditions such as arthritis and metabolic syndrome [23,48,199,200].

-Cardiovascular protection

Anthocyanins are frequently marketed as cardioprotective agents, with observational studies linking high intake (e.g., from berries) to reduced blood pressure, improved lipid profiles, and lower CVD risk. For instance, nurses’ health study II associated higher anthocyanin consumption with a decreased risk of myocardial infarction. However, these findings are largely correlational, and randomized controlled trials (RCTs) show mixed results. Blood pressure and endothelial function, there are some RCTs report modest reductions in systolic blood pressure (∼3–5 mmHg) after berry consumption, but others find no significant effect. These discrepancies may stem from differences in anthocyanin doses, food matrices, and participant baseline health. Lipid metabolism while in vitro studies suggest anthocyanins inhibit cholesterol synthesis, human trials show inconsistent effects on LDL-C and HDL-C, possibly due to poor bioavailability (∼1–2% absorption). Critical issue many studies use whole foods (e.g., blueberries) rather than isolated anthocyanins, making it difficult to attribute effects solely to these compounds. Confounding factors (e.g., fiber, vitamin C) may contribute to the benefits observed [201,202]. Several epidemiological and clinical studies suggest that anthocyanin-rich diets are associated with a reduced risk of cardiovascular disease (CVD). The mechanisms include an improvement in endothelial function; enhanced nitric oxide (NO) production; and the promotion of vasodilation. In addition, a reduction in the oxidation of LDL oxidation prevents atherosclerosis by inhibiting the formation of foam cells. Furthermore, a lower blood pressure modulates ACE (angiotensin-converting enzyme) activity, improves lipid profiles, decreases triglycerides and increases HDL cholesterol [203,204,205].

-Antidiabetic Effects

Anthocyanins improve glucose metabolism by enhancing insulin sensitivity Via AMPK activation and by inhibiting carbohydrate-digesting enzymes (α-amylase, α-glucosidase), reducing postprandial glucose spikes, and protecting pancreatic β-cells from oxidative damage [206,207,208].

-Neuroprotective properties

Anthocyanins exert neuroprotective effects by crossing the blood–brain barrier and reducing neuroinflammation [209,210]. They also inhibit the aggregation of amyloid-β (linked to Alzheimer’s disease) and improve cognitive function by enhancing synaptic plasticity. Anthocyanins are proposed to mitigate neurodegeneration (e.g., Alzheimer’s) by reducing oxidative stress and amyloid-beta aggregation. Small human trials report improved cognitive function with berry intake, but lack of long-term RCTs most studies are short-term or rely on animal models. Blood–brain barrier penetration It is unclear whether anthocyanins or their metabolites reach the brain in sufficient quantities [211].

-Anticancer Potential

Anthocyanins exhibit antiproliferative and pro-apoptotic effects in cancer cell lines, attributed to their antioxidant and anti-inflammatory properties (e.g., inhibition of NF-κB). However, translating these findings to humans remains challenging. Bioavailability issues most in vitro studies use concentrations (10–100 μM) far exceeding physiological levels (<1 μM in plasma). Animal vs. human data while rodent models show tumor suppression (e.g., in colon cancer), human epidemiological data are inconclusive. The European Food Safety Authority (EFSA) has rejected claims that anthocyanins reduce cancer risk due to insufficient evidence [212]. Although research is still evolving, anthocyanins may suppress tumor growth by inducing apoptosis in cancer cells via p53 activation and suppress tumor angiogenesis by inhibiting VEGF, as well as by modulating carcinogen-metabolizing enzymes (e.g., cytochrome P450) [213,214,215]. 

-Weight management

Anthocyanins may aid in the prevention of obesity by reducing adipogenesis (fat cell formation) via the down-regulation of PPAR-γ and enhance thermogenesis via the activation of brown adipose tissue [216,217].

-Gut microbiota modulation

Anthocyanins may serve as prebiotics, promoting beneficial bacteria (e.g., bifidobacterium, lactobacillus), and produce SCFAs (short-chain fatty acids), which improve the integrity of the gut barrier [217,218].

Anthocyanins are potent bioactive compounds with significant health benefits, particularly due to their ability to reduce oxidative stress, inflammation, and the risk of chronic disease. While their bioavailability is a challenge, consuming a diverse diet that is rich in anthocyanin-containing foods can enhance overall health. Further research is needed to establish the optimal doses and long-term effects of anthocyanins in humans. Incorporating berries, purple vegetables, and whole grains into daily nutrition would allow us to harness their benefits. Future research should focus on improving their bioavailability through nano-encapsulation and synergistic food combinations, conducting large-scale human trials to establish dose–response relationships, and exploring personalized nutrition based on gut microbiota profiles. Incorporating anthocyanin-rich foods into a balanced diet is a practical strategy for harnessing their health-promoting effects, as shown in Figure 7 [219] the implications of anthocyanin at human health. While anthocyanins show promise in preclinical research, many health claims exceed current evidence. Their benefits may be real but are likely modest, context-dependent, and influenced by dietary matrix effects. Consumers should be wary of hyperbolic marketing, and researchers must address bioavailability and mechanistic gaps before definitive conclusions can be drawn.

## 5. Conclusions

Anthocyanins, the vibrant pigments responsible for the red, purple, and blue hues in rice and barley, are not only crucial for plant defense and pollination, but also offer significant health benefits for humans. Their occurrence varies among different cultivars, with pigmented varieties of rice (e.g., black and red rice) and barley (e.g., purple barley) containing higher concentrations of these bioactive compounds. The biosynthesis of anthocyanins in these grains is regulated by complex genetic and environmental factors, involving key enzymes such as chalcone synthase (CHS), flavonoid 3-hydroxylase (F3H), and dihydroflavonol 4-reductase (DFR), as well as transcription factors such as MYB and bHLH [76,220,221]. From a nutritional perspective, the anthocyanins present in rice and barley exhibit potent antioxidant, anti-inflammatory, and anti-carcinogenic properties [222,223]. Their consumption has been linked to a reduced risk of chronic diseases, including cardiovascular disorders, diabetes, and certain cancers [224]. Additionally, they can improve our metabolic health and may help mitigate oxidative stress. As research continues to uncover their full potential, anthocyanin-rich rice and barley varieties hold promise as functional foods with the potential to enhance dietary health benefits. Future studies should focus on optimizing cultivation techniques, improving their bioavailability, and further elucidating their mechanisms of action to maximize their therapeutic applications. In summary, the anthocyanins present in rice and barley represent a valuable intersection of agricultural science and human nutrition, offering both esthetic and health-promoting qualities that warrant further exploration and utilization in food and medicine.

## Figures and Tables

**Figure 1 ijms-26-06225-f001:**
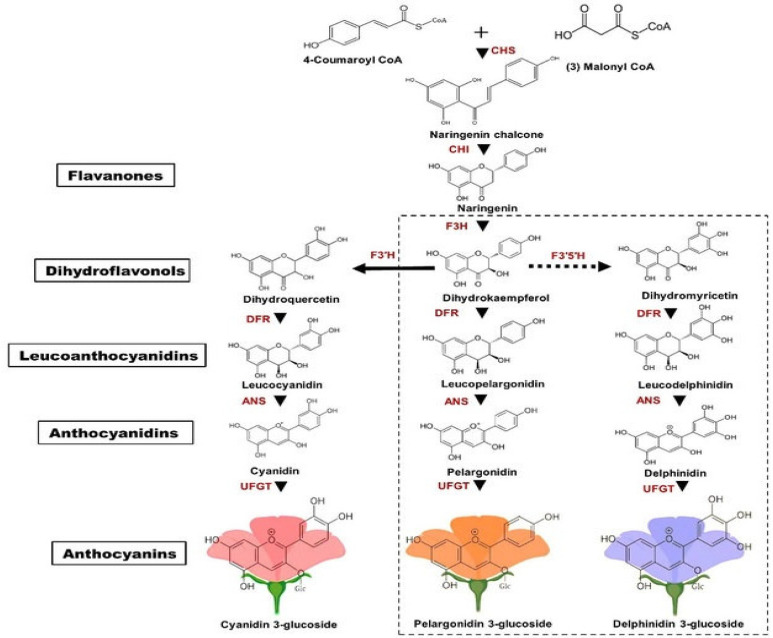
Cyanidin-3-glucoside biosynthesis pathway.

**Figure 2 ijms-26-06225-f002:**
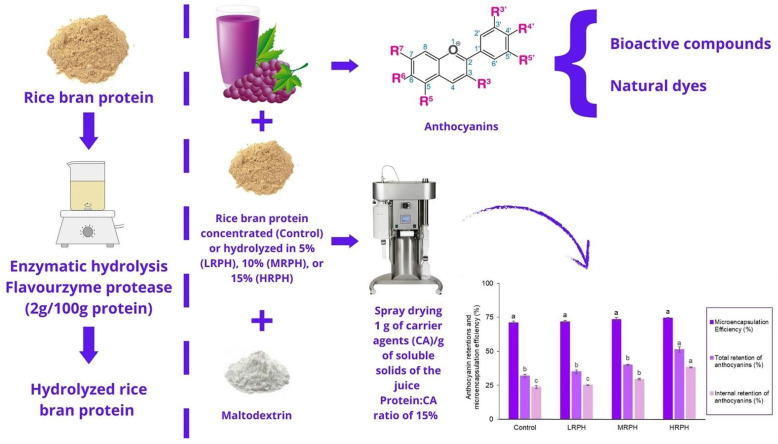
Extraction and qualification of anthocyanins in rice and grapes. While a; grapes, b: rice bran, and c: maltodextrin.

**Figure 3 ijms-26-06225-f003:**
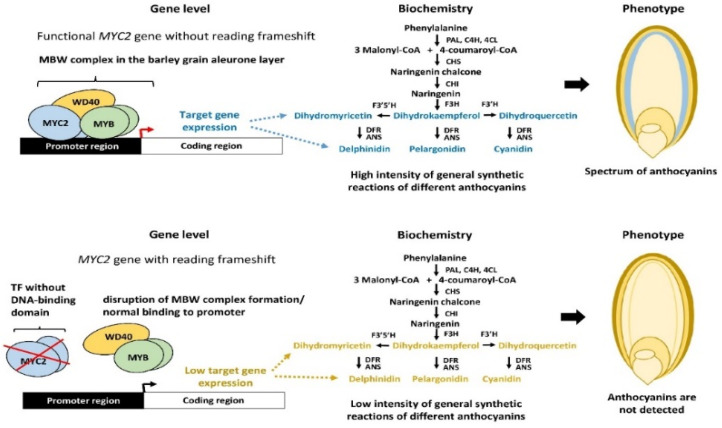
Transcription factors regulate anthocyanin production.

**Figure 4 ijms-26-06225-f004:**
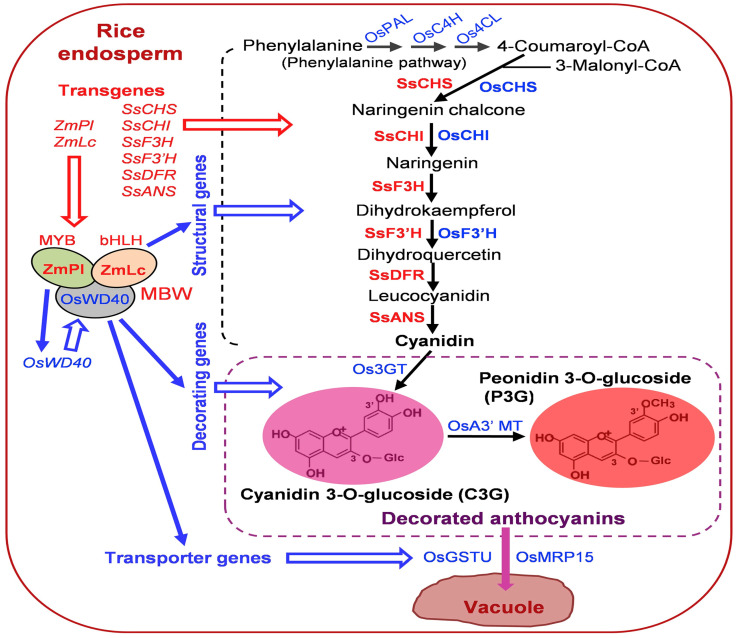
Transcription factors and its role to enhance the anthocyanin content cereal grains.

**Figure 5 ijms-26-06225-f005:**
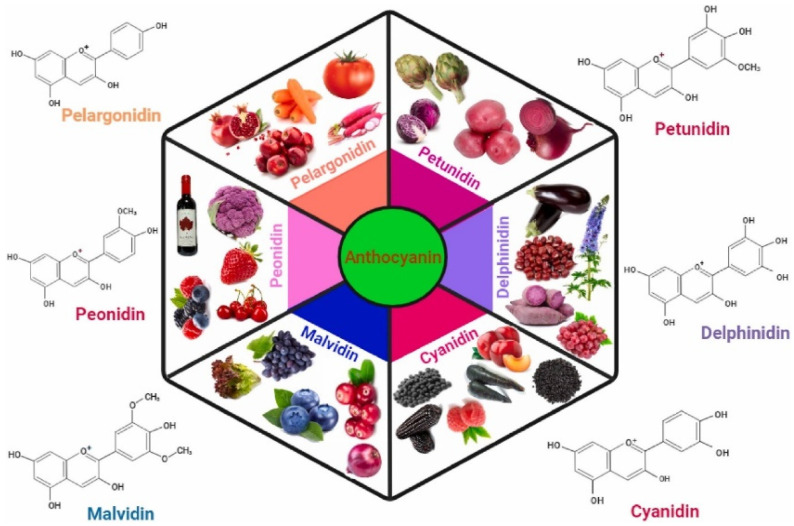
Type of anthocyanins and its biochemical composition.

**Figure 6 ijms-26-06225-f006:**
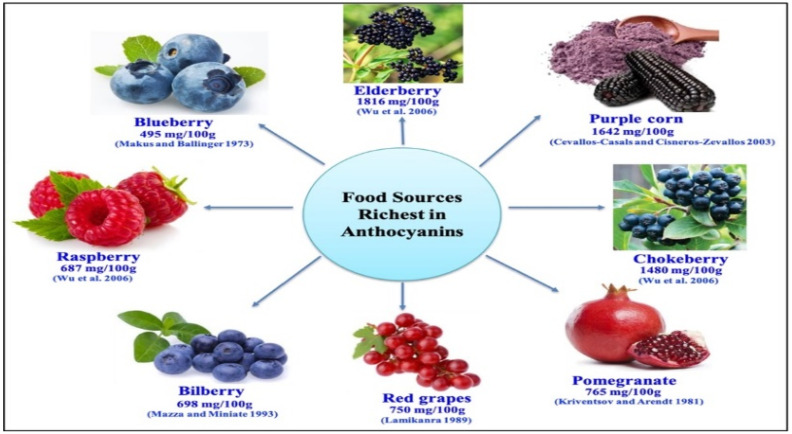
Food sources most rich in anthocyanins [177,178,179,180,181,182].

**Figure 7 ijms-26-06225-f007:**
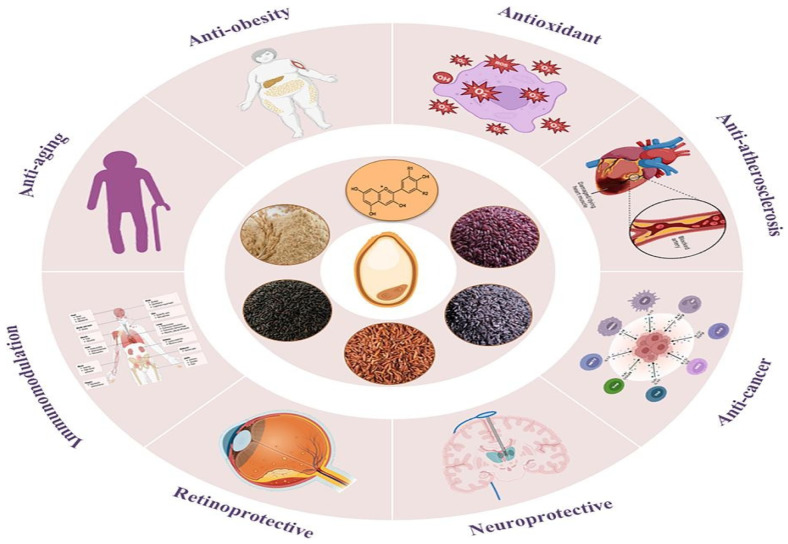
The nutritional implications and health benefits of anthocyanins.

**Table 1 ijms-26-06225-t001:** Analysis of the compositional diversity of anthocyanins (cyanidin-3-glucoside in black rice vs. acylated forms in barley).

Feature	Black Rice (Cyanidin-3-Glucoside, C3G)	Barley (Acylated Anthocyanins)	References
Major anthocyanin type	Non-acylated (simple glycoside)	Acylated (complex forms)	[6]
Primary anthocyanins	Cyanidin-3-glucoside (dominant)	Cyanidin-3-glucoside acylated with phenolic acids (e.g., sinapic, coumaric, ferulic acids)	[31]
Color stability	Less stable (degrades faster under heat/light)	More stable due to acylation (protects against degradation)	[32]
Bioavailability	Higher absorption (simpler structure)	Lower initial absorption (complex structure), but slower metabolism	[32,33]
Health benefits	Strong antioxidant, anti-inflammatory	Enhanced antioxidant capacity due to acyl groups	[33]
Occurrence in grain	Concentrated in the bran layer	Distributed in aleurone/pericarp layers	[31]
Genetic control	Controlled by a few key genes (e.g., *OsANS*, *OsDFR*)	Complex biosynthesis involving acyltransferases (e.g., *HvAT*)	[7]
Environmental influence	Moderate (affected by soil nutrients)	High (acylation influenced by stress conditions)	[7,31]
Processing sensitivity	High (leaching during cooking)	More resistant to processing (stable in baked/fermented products)	[31]

**Table 2 ijms-26-06225-t002:** Anthocyanin composition in rice and barley.

Feature	Rice (Black/Purple)	Barley (Purple/Black)	References
Major anthocyanins	Cyanidin-3-glucoside (C3G), Peonidin-3-glucoside (P3G)	Cyanidin-3-glucoside (C3G), Delphinidin-3-glucoside	[31,47]
Other compounds	Malvidin, Petunidin (minor)	Pelargonidin, Peonidin (minor)	[47]
Pigment location	Primarily in the bran layer (pericarp)	Distributed in the aleurone layer and hull	[32]
Color influence	Deep purple to black	Purple to blue-black	[32]
Concentration	100–500 mg/100 g (varies by cultivar)	50–300 mg/100 g (varies by cultivar)	[31,47]
Health benefits	Antioxidant, anti-inflammatory, cardiovascular support	Antioxidant, anti-diabetic, neuroprotective	[48]
Genetic control	Regulated by transcription factors like *OsC1*, *Rb*	Controlled by *Ant2* and *Ant13* genes	[48,49]
Stability	Sensitive to heat and pH changes	More stable due to matrix interactions in grain	[47]
Common uses	Colored rice dishes, supplements, natural dye	Functional foods, brewing, flour fortification	[48,49]

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
