# Peer review of "Occurrence, Biosynthesis, and Health Benefits of Anthocyanins in Rice and Barley"

_ijms, 2025, doi:10.3390/ijms26136225_

Round 1
Reviewer 1 Report
Comments and Suggestions for Authors
Review on manuscript: IJMS_3701566_R1
‘Occurrence, Biosynthesis, and Health Benefits of Anthocyanins in Rice and Barley’
by Essam A. ElShamey 1,2*, Xiaomeng Yang 1, Jiazhen Yang 1, Xiaoying Pu 1, Li´E Yang 1, Changjiao Ke 1 and Zeng Yawen1
submitted to International Journal of Molecular Sciences
Decision: Minor Revision
Comments and Suggestions for Authors
The paper ‘Occurrence, Biosynthesis, and Health Benefits of Anthocyanins in Rice and Barley’ is an interesting review that reports many studies, even recent ones, that concern the genetic and molecular bases of the anthocyanins biosynthesis in rice and barley in order to optimize strategies to improve nutritional value, positioning them as valuable functional foods in the global market.
In my opinion, this paper is of interests and it fits in the scope of International Journal of Molecular Sciences, however, there are some corrections to do along the manuscript. I believe that the article can be improved and after that suitable for publication.
Please note that from my MDPI account I am only allowed to download the pdf version without line numbers, so to indicate my comments I will specify paragraph and page number, I hope they are clear
My comments are as follows:
In the title of paragraph 2 on the second page of the manuscript please remove the bracket with eg, it is not necessary.
The sentence ‘This review synthesizes recent advancements in anthocyanin analysis in rice and barley, focusing on extraction methods, quantification techniques, and bioactive potential.’ is fine for the introduction or for the abstract, I suggest moving it from paragraph 2.
In the text of paragraph 2 I suggest adding a sentence that better explains table 1, for example: table 1 reports the studies on...
Table 1: please remove 'e.g.' from the table title, I also recommend indicating the articles in full in the reference column in order to better understand which points of the table have been reported in which studies.
Fig.2 : I found this figure as Graphical abstract of Almeida, R. F., Gomes, M. H. G., & Kurozawa, L. E. (2024). Enzymatic hydrolysis improves the encapsulation properties of rice bran protein by increasing retention of anthocyanins in microparticles of grape juice. Food Research International, 180, 114090.
I am not convinced that it is correct to use this figure without explaining that in this case the anthocyanins are derived from grape juice, it is better to explain better and in any case cite the authors of the study.
Table 2: please remove 'for example' from the text, as for table 1 I suggest to specify the articles in full in the reference column in order to better understand which points of the table have been reported in which studies.
At 2.2.1.2. paragraph please change ‘Transcription factors are regulated anthocyanin production’ in Transcription factors regulate anthocyanin production’
Fig.3 : I found this figure as Graphical abstract of ‘Egorova, A. A., Zykova, T. E., Hertig, C. W., Hoffie, I., Morozov, S. V., Chernyak, E. I., ... & Khlestkina, E. K. (2024). Accumulation of Anthocyanin in the Aleurone of Barley Grains by Targeted Restoration of the MYC2 Gene. International Journal of Molecular Sciences,25(23), 12705.’ Please check if it is the correct figure according to what is described in the text, I have some doubts about the transcription factor mentioned, and if you keep the figure please cite the correct authors in the caption and in the references.
Fig.4 : I found this figure in ‘Zhu, Q.; Yu, S.; Zeng, D.; Liu, H.; Wang, H.; Yang, Z.; Xie, X.; Shen, R.; Tan, J.; Li, H. Development of “purple endosperm rice” by engineering anthocyanin biosynthesis in the endosperm with a high-efficiency transgene stacking system. Molecular plant 2017, 10, 918-929.’ Ref [43]. Please cite the correct authors in the text and correct the caption of figure ‘Engineering of the anthocyanin biosynthesis pathway in the rice endosperm.’ as described by the authors from whose article the figure was taken.
IN GENERAL PLEASE DO NOT USE PARAGRAPH TITLES FOR FIGURE CAPTIONS, THEY ARE TOO GENERIC COMPARED TO WHAT IS DESCRIBED BY THE FIGURES THEMSELVES.
Fig.5 : I found this figure in ‘Lakshmikanthan, M., Muthu, S., Krishnan, K., Altemimi, A. B., Haider, N. N., Govindan, L., ... & Francis, Y. M. (2024). A comprehensive review on anthocyanin-rich foods: Insights into extraction, medicinal potential, and sustainable applications. Journal of Agriculture and Food Research, 101245..’ as before I can't find any feedback to the article in the references, furthermore the original caption states that the image was created through the site biorender (Source from ww.biorender.com) check carefully that the image can be used and that there are no copyrights and if you use it cite the correct authors in the references.
At 3 paragraph the following sentence is a repetition, it can be removed ‘are belonging to the flavonoid family, responsible for the red, purple, and blue colours in many plants, including certain varieties of rice (e.g., black rice) and barley (e.g., purple barley).’
Table 3: as for table 1and 2 I suggest to specify the articles in full in the reference column in order to better understand which points of the table have been reported in which studies.
At 3.7. paragraph Strategies to enhance anthocyanin stability please specify the correct references for each list item
Fig.7 : I found this figure as Graphical abstract of ‘Chen, T., Xie, L., Wang, G., Jiao, J., Zhao, J., Yu, Q., ... & Xie, J. (2024). Anthocyanins-natural pigment of colored rice bran: Composition and biological activities. Food Research International,175, 113722.’ please cite the correct authors in the caption and in the references.
In conclusion I would like to underline that the review can be published only when the correct citations are inserted.
Author Response
Dear Editor,
I would like to resubmit our paper “Occurrence, Biosynthesis, and Health Benefits of Anthocyanins in Rice and Barley” Manuscript ID: ijms-3701566 it was pending for major revisions for publication, and English language was improved by native MDPI- Author Service (English-Editing-Certificate-95776), and please let me present and clear some points for reply at the respected referees:
For Reviewer 1
Reviewer suggest “In the title of paragraph 2 on the second page of the manuscript please remove the bracket with eg, it is not necessary”:
Authors Reply:
Thank you for pointing this out. I/We agree with this comment. It was done at line 65, 66.
-------------------------------------------------------------------------------------------------------------------
Reviewer suggest “The sentence ‘This review synthesizes recent advancements in anthocyanin analysis in rice and barley, focusing on extraction methods, quantification techniques, and bioactive potential.’ is fine for the introduction or for the abstract, I suggest moving it from paragraph 2”.
Authors Reply:
Thank you for pointing this out. I/We agree with this comment. It was done and removed at line 77 – 79.
----------------------------------------------------------------------------------------------------------------
Reviewer suggest “In the text of paragraph 2 I suggest adding a sentence that better explains table 1, for example: table 1 reports the studies on…
Authors Reply:
Thank you for pointing this out. I/We agree with this comment. It was done and added at line 74 - 75
---------------------------------------------------------------------------------------------------------------------------
Reviewer suggest “Table 1: please remove 'e.g.' from the table title, I also recommend indicating the articles in full in the reference column in order to better understand which points of the table have been reported in which studies.”.
Authors Reply:
Thank you for pointing this out. I/We agree with this comment. It was done and removed at line 82, and references were added to the Table separately.
--------------------------------------------------------------------------------------------------------------------------
Reviewer suggest “Fig.2 : I found this figure as Graphical abstract of Almeida, R. F., Gomes, M. H. G., & Kurozawa, L. E. (2024). Enzymatic hydrolysis improves the encapsulation properties of rice bran protein by increasing retention of anthocyanins in microparticles of grape juice. Food Research International, 180, 114090.
I am not convinced that it is correct to use this figure without explaining that in this case the anthocyanins are derived from grape juice, it is better to explain better and, in any case, cite the authors of the study”:
Authors Reply:
Thank you for pointing this out. I/We agree with this comment. It was done and added at line 98, and for citation was added at line 85, and the image can be used and there are no copyrights.
-------------------------------------------------------------------------------------------------------------------
Reviewer suggest “Table 2: please remove 'for example' from the text, as for table 1 I suggest to specify the articles in full in the reference column in order to better understand which points of the table have been reported in which studies”.
Authors Reply:
Thank you for pointing this out. I/We agree with this comment. It was done and removed at line 111, and references were added to the Table separately.
----------------------------------------------------------------------------------------------------------------
Reviewer suggest “At 2.2.1.2. paragraph please change ‘Transcription factors are regulated anthocyanin production’ in Transcription factors regulate anthocyanin production’
Authors Reply:
Thank you for pointing this out. I/We agree with this comment. It was done at line 164
---------------------------------------------------------------------------------------------------------------------------
Reviewer suggest “Fig.3 : I found this figure as Graphical abstract of ‘Egorova, A. A., Zykova, T. E., Hertig, C. W., Hoffie, I., Morozov, S. V., Chernyak, E. I., ... & Khlestkina, E. K. (2024). Accumulation of Anthocyanin in the Aleurone of Barley Grains by Targeted Restoration of the MYC2 Gene. International Journal of Molecular Sciences,25(23), 12705.’ Please check if it is the correct figure according to what is described in the text, I have some doubts about the transcription factor mentioned, and if you keep the figure, please cite the correct authors in the caption and in the references.”.
Authors Reply:
Thank you for pointing this out. I/We agree with this comment. It was done and a citation was added at line 168, and the image can be used and there are no copyrights.
----------------------------------------------------------------------------------------------------------------
Reviewer suggest “Fig.4 : I found this figure in ‘Zhu, Q.; Yu, S.; Zeng, D.; Liu, H.; Wang, H.; Yang, Z.; Xie, X.; Shen, R.; Tan, J.; Li, H. Development of “purple endosperm rice” by engineering anthocyanin biosynthesis in the endosperm with a high-efficiency transgene stacking system. Molecular plant 2017, 10, 918-929.’ Ref [43]. Please cite the correct authors in the text and correct the caption of figure ‘Engineering of the anthocyanin biosynthesis pathway in the rice endosperm.’ as described by the authors from whose article the figure was taken.’
Authors Reply:
Thank you for pointing this out. I/We agree with this comment. It was done and a citation was added at line 190, and the image can be used and there are no copyrights.
-------------------------------------------------------------------------------------------------------------------------
Reviewer suggest “Fig.5 : I found this figure in ‘Lakshmikanthan, M., Muthu, S., Krishnan, K., Altemimi, A. B., Haider, N. N., Govindan, L., ... & Francis, Y. M. (2024). A comprehensive review on anthocyanin-rich foods: Insights into extraction, medicinal potential, and sustainable applications. Journal of Agriculture and Food Research, 101245..’ as before I can't find any feedback to the article in the references, furthermore the original caption states that the image was created through the site biorender (Source from ww.biorender.com) check carefully that the image can be used and that there are no copyrights and if you use it cite the correct authors in the references.’
Authors Reply:
Thank you for pointing this out. I/We agree with this comment. It was done and a citation was added at line 321, and the image can be used and there are no copyrights.
-------------------------------------------------------------------------------------------------------------------------
Reviewer suggest “At 3 paragraph the following sentence is a repetition, it can be removed ‘are belonging to the flavonoid family, responsible for the red, purple, and blue colours in many plants, including certain varieties of rice (e.g., black rice) and barley (e.g., purple barley).’’
Authors Reply:
Thank you for pointing this out. I/We agree with this comment. It was done and removed at line 375-377.
-------------------------------------------------------------------------------------------------------------------------
Reviewer suggest “Table 3: as for table 1and 2 I suggest to specify the articles in full in the reference column in order to better understand which points of the table have been reported in which studies.’
Authors Reply:
Thank you for pointing this out. I/We agree with this comment. It was done and references were added to the Table separately.
-------------------------------------------------------------------------------------------------------------------------
Reviewer suggest “At 3.7. paragraph Strategies to enhance anthocyanin stability please specify the correct references for each list item’
Authors Reply:
Thank you for pointing this out. I/We agree with this comment. It was done and a citation was added at lines 510, 512, 514, and 516.
-------------------------------------------------------------------------------------------------------------------------
Reviewer suggest “Fig.7 : I found this figure as Graphical abstract of ‘Chen, T., Xie, L., Wang, G., Jiao, J., Zhao, J., Yu, Q., ... & Xie, J. (2024). Anthocyanins-natural pigment of colored rice bran: Composition and biological activities. Food Research International,175, 113722.’ please cite the correct authors in the caption and in the references.’
Authors Reply:
Thank you for pointing this out. I/We agree with this comment. It was done and a citation was added at line 648, and the image can be used and there are no copyrights.
-------------------------------------------------------------------------------------------------------------------------
Finally, thank you for your suggestions, they added value to our review. We tried our best to improve the manuscript. We appreciate the reviewer’s warm work earnestly and hope the correction will meet with approval. Once again, thank you very much for your comments and suggestions.
Thank you for taking the time to consider our paper. We hope you find it suitable for publishing.
Yours sincerely,
Prof. Zeng Yawen
Biotechnology and Germplasm Resources Research Institute,
Yunnan Academy of Agricultural Sciences, Kunming, China
Reviewer 2 Report
Comments and Suggestions for Authors
Dear Authors,
Thank you for submitting your manuscript titled “Occurrence, Biosynthesis, and Health Benefits of Anthocyanins in Rice and Barley”. Your paper presents a comprehensive overview of anthocyanins in cereal crops, integrating biosynthesis pathways, genetic regulation, environmental factors, and their nutritional significance.
While your manuscript offers considerable value, I have identified several areas needing improvement:
-
The writing contains multiple grammatical errors and awkward constructions that hinder clarity.
-
Certain captions are incomplete or misleading (e.g., Figure 3 and 7).
-
Some sections are overly redundant and can be better focused.
-
The manuscript would benefit from deeper critical discussion, especially regarding health claims and genetic engineering challenges.
I recommend a major revision with a strong emphasis on improving language, tightening structure, and expanding analytical depth. Please also ensure proper figure integration and review references for completeness.
I appreciate the efforts put into compiling this work and look forward to reviewing the revised version.
Sincerely,
Comments on the Quality of English LanguageDear Editor,
I have reviewed the manuscript titled “Occurrence, Biosynthesis, and Health Benefits of Anthocyanins in Rice and Barley” submitted to the International Journal of Molecular Sciences. The article addresses a relevant and increasingly important topic in food science and molecular nutrition.
While the manuscript demonstrates thorough research and integrates graphical elements well, it is currently hindered by substantial grammatical and structural issues. In addition, the discussion lacks critical evaluation in several sections.
I recommend major revision at this stage. With substantial improvement in language, figure caption clarity, and reduction of redundancy, the paper could become a strong candidate for publication.
Best regards,
Please see my comments below:
This manuscript provides a thorough and multidisciplinary review of anthocyanins in Oryza sativa (rice) and Hordeum vulgare (barley), covering occurrence, biosynthesis, compositional diversity, environmental factors, genetic regulation, and health benefits. The topic is timely and relevant to plant scientists, nutritionists, and food technologists. The authors have done well to integrate biochemical, genetic, and health-related perspectives, and the graphical content is visually engaging.
However, the paper suffers from substantial grammatical errors, inconsistencies in writing style, improper or awkward phrasing, and figure caption issues. These linguistic problems significantly reduce the manuscript's readability and professional polish. Furthermore, the manuscript would benefit from a more critical and less repetitive discussion, with a clearer delineation between evidence-based facts and speculative content.
2. Major Strengths
Comprehensive Scope: Covers anthocyanin biosynthesis, genetic regulation, environmental modulation, food processing implications, and human health effects.
Visual Aids: Seven figures effectively illustrate pathways, extraction methods, regulation, and nutritional roles.
Current References: Cites very recent literature (up to 2025), indicating the authors' awareness of ongoing research.
Functional Food Relevance: Strong link between molecular science and potential application in human health.
3. Major Issues
A. Language and Grammar
The manuscript contains numerous grammatical errors and awkward phrases. Examples:
“Transcription factors are regulated anthocyanin production” → Should be: “Transcription factors regulating anthocyanin production”.
“The effect of water stress on rice as fellow” → Should be: “The effect of water stress on rice is as follows”.
Sentence construction often lacks clarity and natural academic flow. A professional English editing service is highly recommended.
B. Repetition and Redundancy
Many sections, especially on genetic regulation, bioavailability, and stability, reiterate similar points without advancing the discussion. E.g., overuse of "Anthocyanins in rice and barley..." in different sections.
The health effects are described in several overlapping ways (e.g., antioxidant role discussed in abstract, introduction, conclusion, and section 4.3).
C. Lack of Critical Analysis
While the article compiles data well, it lacks critical appraisal. For example:
Health benefits of anthocyanins are extensively discussed, but with minimal distinction between in vitro, animal, and clinical human evidence.
Genetic engineering is promoted, but challenges like consumer acceptance, regulatory frameworks, and unintended consequences are not adequately discussed.
D. Figure Captions and Consistency
Some figure captions are grammatically incorrect or incomplete:
Figure 7 caption ends abruptly: “Anthocyanins nutritional implications and its health benefits and”.
Figure 3 has a structural error.
Figures should be referred to in the main text at appropriate places and interpreted more explicitly.
E. Structural and Formatting Issues
Section numbering is inconsistent. For example, subheadings under section 2 do not follow a clean hierarchy.
Long, unbroken paragraphs (e.g., introduction) can be split into sub-topics for better readability.
4. Minor Issues
Several factual claims are not directly supported by citations (e.g., specific concentration ranges, post-harvest losses).
Acronyms (e.g., C3G, TFs, MBW) should be consistently introduced at first use.
The term “as fellow” repeatedly appears instead of “as follows”—a possible auto-translation or copy-editing error.
Typographic and style inconsistencies: some headers are bold, others italic, some use full justification while others are not aligned.
5. Suggestions for Improvement
Professional proofreading and English editing is essential.
Streamline redundant content and focus each section more sharply.
Add tables summarizing clinical findings, environmental effects, and genetic variants for clarity.
Include a graphical abstract or summary diagram showing the integrated picture of biosynthesis, enhancement, and health impacts.
Where possible, add quantitative data to support qualitative statements (e.g., average increases due to genetic engineering or light exposure).
6. Final Recommendation
Major Revision
This manuscript has the potential to be an impactful and highly cited review, especially in the field of functional cereals and nutritional genomics. However, major linguistic, editorial, and structural improvements are necessary before it is suitable for publication.
Author Response
Dear Editor,
I would like to resubmit our paper “Occurrence, Biosynthesis, and Health Benefits of Anthocyanins in Rice and Barley” Manuscript ID: ijms-3701566 it was pending for major revisions for publication, and English language was improved by native MDPI- Author Service (English-Editing-Certificate-95776), and please let me present and clear some points for reply at the respected referees:
For Reviewer 2
Reviewer suggest “Thank you for submitting your manuscript titled “Occurrence, Biosynthesis, and Health Benefits of Anthocyanins in Rice and Barley”. Your paper presents a comprehensive overview of anthocyanins in cereal crops, integrating biosynthesis pathways, genetic regulation, environmental factors, and their nutritional significance.
While your manuscript offers considerable value, I have identified several areas needing improvement:
- The writing contains multiple grammatical errors and awkward constructions that hinder clarity.
- Certain captions are incomplete or misleading (e.g., Figure 3 and 7).
- Some sections are overly redundant and can be better focused.
- The manuscript would benefit from deeper critical discussion, especially regarding health claims and genetic engineering challenges
Authors Reply:
- For point 1. Thank you for pointing this out. I/We agree with this comment. It was done by MDPI-Author Service (English-Editing-Certificate-95776)
- For point 2. Thank you for pointing this out. I/We agree with this comment. It was improved and done at line 168-169 and 655
- For point 3. Thank you for pointing this out. I/We agree with this comment. It was improved and done by MDPI-Author Service (English-Editing-Certificate-95776) for grammatical errors, and some sections were improved, and can find at lines 129-136, 147-164, 173-182, 191-214, 220-252, 286-314, 316-318, 375-377, 563-570, 581-594, 611-625, 648-653.
- For point 4. Thank you for pointing this out. I/We agree with this comment. It was done and improved deep discussion at line 220-252, 286-314 for challenges in genetic engineering, and for health claims it was done, and discussion was improved at line 563-570, 581-594, 611-616, 618-625, and 648-653.
---------------------------------------------------------------------------------------------------------------------
Reviewer suggest “This manuscript provides a thorough and multidisciplinary review of anthocyanins in Oryza sativa (rice) and Hordeum vulgare (barley), covering occurrence, biosynthesis, compositional diversity, environmental factors, genetic regulation, and health benefits. The topic is timely and relevant to plant scientists, nutritionists, and food technologists. The authors have done well to integrate biochemical, genetic, and health-related perspectives, and the graphical content is visually engaging.
However, the paper suffers from substantial grammatical errors, inconsistencies in writing style, improper or awkward phrasing, and figure caption issues. These linguistic problems significantly reduce the manuscript's readability and professional polish. Furthermore, the manuscript would benefit from a more critical and less repetitive discussion, with a clearer delineation between evidence-based facts and speculative content”.
Authors Reply:
Thank you for pointing this out. I/We agree with this comment. It was done as mentioned above.
----------------------------------------------------------------------------------------------------------------
Reviewer suggest “A. Language and Grammar
The manuscript contains numerous grammatical errors and awkward phrases. Examples:
“Transcription factors are regulated anthocyanin production” → Should be: “Transcription factors regulating anthocyanin production”.
“The effect of water stress on rice as fellow” → Should be: “The effect of water stress on rice is as follows”.
Sentence construction often lacks clarity and natural academic flow. A professional English editing service is highly recommended.”:
Authors Reply:
Thank you for pointing this out. I/We agree with this comment. It was done by MDPI-Author Service (English-Editing-Certificate-95776)
---------------------------------------------------------------------------------------------------------------------------
Reviewer suggest “B. Repetition and Redundancy
- Many sections, especially on genetic regulation, bioavailability, and stability, reiterate similar points without advancing the discussion. E.g., overuse of "Anthocyanins in rice and barley..." in different sections.
- The health effects are described in several overlapping ways (e.g., antioxidant role discussed in abstract, introduction, conclusion, and section 4.3)”.
Author Reply:
- For point 1. Thank you for pointing this out. I/We agree with this comment. It was done and improved to lines 220-252, 286-314
- For point 2. Thank you for pointing this out. I/We agree with this comment. It was done at line 563-570, 581-594, 611-616, 618-625, and 648-653.
----------------------------------------------------------------------------------------------------------------
Reviewer suggest “C. Lack of Critical Analysis While the article compiles data well, it lacks critical appraisal. For example:
- Health benefits of anthocyanins are extensively discussed, but with minimal distinction between in vitro, animal, and clinical human evidence.
- Genetic engineering is promoted, but challenges like consumer acceptance, regulatory frameworks, and unintended consequences are not adequately discussed.
Authors Reply:
Author Reply:
- For point 1. Thank you for pointing this out. I/We agree with this comment. It was done at line 563-570, 581-594, 611-616, 618-625, and 648-653.
- For point 2. Thank you for pointing this out. I/We agree with this comment. It was done and improved at line 220-252, 286-314.
---------------------------------------------------------------------------------------------------------------------------
Reviewer suggest “D. Figure Captions and Consistency Some figure captions are grammatically incorrect or incomplete:
- Figure 7 caption ends abruptly: “Anthocyanins nutritional implications and its health benefits and”.
- Figure 3 has a structural error.
- Figures should be referred to in the main text at appropriate places and interpreted more explicitly.
Authors Reply:
- For point 1. Thank you for pointing this out. I/We agree with this comment. It was done at line 655.
- For point 2. Thank you for pointing this out. I/We agree with this comment. It was done at line 168-171.
- For point 3. Thank you for pointing this out. I/We agree with this comment. It was done at lines 69, 85, 168-169, 189, 320-321, 535 and 648.
---------------------------------------------------------------------------------------------------------------------------
Reviewer suggest “E. Structural and Formatting Issues
- Section numbering is inconsistent. For example, subheadings under section 2 do not follow a clean hierarchy.
- Long, unbroken paragraphs (e.g., introduction) can be split into sub-topics for better readability.”:
Authors Reply:
- For point 1. Thank you for pointing this out. I/We agree with this comment. It was done at lines 254-265, 266-270, 271
- For point 2. Thank you for pointing this out. I/We agree with this comment. It was done at line 56.
---------------------------------------------------------------------------------------------------------------------------
Reviewer suggest “4. Minor Issues
- Several factual claims are not directly supported by citations (e.g., specific concentration ranges, post-harvest losses).
- Acronyms (e.g., C3G, TFs, MBW) should be consistently introduced at first use.
- The term “as fellow” repeatedly appears instead of “as follows”—a possible auto-translation or copy-editing error.
- Typographic and style inconsistencies: some headers are bold, others italic, some use full justification while others are not aligned.
Authors Reply:
Thank you for pointing this out. I/We agree with this comment. It was done at whole manuscript.
------------------------------------------------------------------------------------------------------------------
Reviewer suggest “5. Suggestions for Improvement
- Professional proofreading and English editing is essential.
- Streamline redundant content and focus each section more sharply.
Authors Reply:
- For point 1. Thank you for pointing this out. I/We agree with this comment. It was done by MDPI-Author Service (English-Editing-Certificate-95776).
- For point 2. Thank you for pointing this out. I/We agree with this comment. It was done and improved at lines 129-136, 147-164, 173-182, 191-214, 220-252, 286-314, 316-318, 375-377, 563-570, 581-594, 611-625, 648-653.
-----------------------------------------------------------------------------------------------------------------------------
Finally, thank you for your suggestions, they added value to our review. We tried our best to improve the manuscript. We appreciate the reviewer’s warm work earnestly and hope the correction will meet with approval. Once again, thank you very much for your comments and suggestions.
Thank you for taking the time to consider our paper. We hope you find it suitable for publishing.
Yours sincerely,
Prof. Zeng Yawen
Biotechnology and Germplasm Resources Research Institute,
Yunnan Academy of Agricultural Sciences, Kunming, China